# Network analysis of depression symptoms and physical activity levels before and after menopause

**Song Chen¹, Di Wu¹, KaiYue Nie², Yuqi Tian³, Ruixin Ma³, Fashui Gao¹, Guofang Ma**[ID]1,3*

**1** Department of Health Management, Xinjiang Medical University, Urumqi, China, **2** Department of Foreign Languages, Xinjiang University, Urumqi, China, **3** Department of Public Health, Xinjiang Medical University, Urumqi, China

\* 420194943@qq.com

## Abstract

### Background

Previous research has established connections between pre- and postmenopause, physical activity, and depression. This study aims to delve deeper into the network structure of depressive symptoms and specific manifestations of these symptoms at different levels of physical activity during pre- and postmenopause, utilizing network analysis as a tool.

### Methods

Our research utilized data samples from the National Health and Nutrition Examination Survey (NHANES) spanning from 2009 to 2018. We assessed depression symptoms through the Patient Health Questionnaire-9, while categorizing physical activity based on the Metabolic Equivalent of Task (MET) values recommended by NHANES and the U.S. physical activity guidelines. We conducted an analysis of the depression symptoms network across varying levels of physical activity, both pre and post-menopause, to identify core symptoms within the network using 'strength' statistics. Furthermore, we evaluated the stability of the network structure via network stability and edge weight difference tests.

### Results

Within the network model of depressive symptoms, both pre- and post-menopause, 'Sad Mood' emerged as the most central symptom, positioning itself as the core of the network. Furthermore, there was a noticeable decrease in the correlation between depressive symptoms and a reduced stability in the network structure during periods of high physical activity compared to those of low physical activity (88.9%→66.7%, 80.5%→72.2%). Notably, no significant structural differences

**Data availability statement:** No additional data are available. NHANES data are publicly available at: URL: https://www.cdc.gov/nchs/nhanes/index.htm. DOI for NHANES metadata: 10.15620/cdc:10640.

**Funding:** The author(s) received no specific funding for this work.

**Competing interests:** The authors have declared that no competing interests exist.

were observed between the pre-menopausal and post-menopausal network models, regardless of physical activity levels ($P_S > 0.05$, $P_M > 0.05$).

## Conclusion

The symptom of 'Sad Mood' is pivotal in the network of depressive symptoms observed in both pre- and post-menopausal women. Engaging in high levels of physical activity may diminish the centrality of this symptom within the network, thereby weakening its association with other symptoms. Prioritizing attention to 'Sad Mood' symptoms during the pre- and post-menopausal phases could be instrumental in mitigating and forestalling the exacerbation of depressive distress.

## Introduction

The World Health Organisation (WHO) characterizes depression as a severe mental health condition, primarily marked by persistent symptoms of sadness, loss of interest, fatigue, and low energy for a minimum of two weeks. Furthermore, depression may manifest concomitantly with anxiety, sleep disturbances, appetite changes, concentration difficulties, guilt feelings, low self-esteem, or suicidal thoughts [1]. Research has indicated that the prevalence of depression is twice as high in women as in men [2], a phenomenon that may be attributed to societal differences in gender roles [3]. Particularly for women approaching menopause, a range of physical and psychological challenges may be experienced due to declining ovarian function and changing hormone levels [4]. These symptoms may include vasodilatory symptoms, sleep problems, mood swings (e.g., depression and anxiety), hair loss, thinning and drying of the skin, weight gain or loss, loss of libido, and vaginal dryness. It is important to note that these changes do not only occur in women with a history of depression, but even those who have not previously experienced such conditions may be affected [5–7] The psychological adjustment required of postmenopausal women further increases their likelihood of developing depressive symptoms and feeling isolated. Consequently, it is imperative to acknowledge the distinctive physiological and psychological characteristics of this demographic when formulating effective interventions [8].

Physical activity (PA) is a health-related behavior that has been extensively researched for its potential to enhance health status and quality of life. It is also recognized as an effective behavioral intervention for treating depression [9]. Multiple studies have demonstrated that PA positively impacts depression prevention and alleviates depressive symptoms across various age groups [10,11]. The mechanisms through which PA mitigates depressive symptoms encompass both neural aspects, such as the release and enhanced efficacy of dopamine, and social dimensions, wherein participation in PA augments feelings of mastery and coherence [12,13]. Prior research has consistently identified a correlation between PA and depressive symptoms [14]. Specifically, variations in PA levels have been linked to differences in depression severity, and exercise has been shown to significantly diminish

depressive symptoms in middle-aged and older women [15]. Moreover, increased levels of PA among perimenopausal women have correlated with reduced psychiatric symptoms [16].

Traditional assessment methods for mental disorders predominantly depend on the compilation of scores from questionnaires, where total scores are derived from an individual's responses to assorted symptoms. This method operates under the assumption that all symptoms stem from the same root cause [17], such as depression leading to a low mood or dietary changes. However, it overlooks the diversity, intricacy, and multifaceted nature of mental health conditions [18]. In contrast, the emerging causal systems perspective on mental disorders posits that clusters of comorbid symptoms like those of depression are not primarily caused by a common source but rather by direct symptom-to-symptom relationships. For instance, low mood, anhedonia, and sleep disturbances are not necessarily outcomes of depression; instead, they may interact via distinct biological and psychological pathways [19,20]. Network analyses provide a comprehensive perspective on the interrelations among symptoms by conceptualizing them as interconnected nodes within a broader network. This approach enables the detailed examination of each symptom's unique contribution and their collective interactions [21]. This perspective allows for an examination of the unique contribution and interaction of each symptom, aiding in the identification of core symptoms integral to disease perpetuation and enabling the creation of personalized treatment plans that target these primary symptoms [22].

In recent years, network analysis methods have been increasingly employed in research on menopausal mental health, offering a novel approach to understanding the intricate interactions among symptoms. Martin-Key et al [23]. identified core sociopsychological symptoms in menopausal women, such as "low mood" and "fatigue," and their associations with initial symptom manifestations using a symptom network model. Wen J et al [24]. further developed a comorbidity network for menopause, anxiety, and depression symptoms, revealing that psychological factors like fatigue and menopausal symptoms like hot flashes, headaches, and dizziness are strongly linked to depression. Moreover, the depression-anxiety statistical network model advanced by Cai et al [25]. confirmed the modular structure of the symptom group, thereby reducing heterogeneity across studies. Notably, these studies did not incorporate physical activity levels as network variables, which restricts their ability to pinpoint behavioral intervention targets. While existing research has associated physical activity with varying degrees of depression severity [26], systematic reviews by Fausto et al [27]. have corroborated the overall benefits of physical activity on menopausal mental health. However, these reviews have not delineated the interactive pathways between exercise behavior and specific symptoms. Thus, this study seeks to elucidate the relationship between physical activity levels during and after menopause and depressive symptoms.

## Methods

### Data source and study population screening

The study population was derived from cross-sectional data from the National Health and Nutrition Examination Survey (NHANES), a nationally representative cross-sectional survey conducted in the United States of America for ten consecutive years from 2009 to 2018 using a multistage probability sampling design by the National Center for Health Statistics (NCHS), a division of the Centers for Disease Control and Prevention (CDC). It included participant data collected through questionnaire interviews, health screenings, and laboratory tests, and the survey was approved by the Ethics Review Board, with all participants providing written informed consent (https://www.cdc.gov/nchs/nhanes/index.htm).

All female participants over the age of 20 years with complete physical activity data and PHQ-9 data were included in this study. Participants with missing information about hysterectomy, bilateral salpingo-oophorectomy, pregnancy, or menstrual status were excluded. Finally, 3755 women were included in the analysis. The flowchart of sample selection is shown in S1 Fig.

Classification of menopausal status was based on the subjects' reproductive health questionnaire, "Have you had at least one menstrual period within the past 12 months?" and "What was the reason for not having a menstrual period within the past 12 months?" with postmenopausal women answering no menstrual periods in the past 12 months and the reason

being menopause or hysterectomy; premenopausal women having menstrual periods in the past 12 months and being younger than 55 years [28].

## Assessment of depressive symptoms

Depression was measured using the Patient Health Questionnaire (PHQ-9), which is composed of nine items. The questionnaire measures various cognitive, affective, physiological, and interpersonal symptoms of depression during the previous two weeks, such as anhedonia, sad mood, sleep, energy, appetite, guilt, concentration, movement, and suicidal ideation. A Likert four-point scale was used to record responses: 0 (not at all), 1 (a few days), 2 (more than half of the days), and 3 (nearly every day). Higher scores represent more severe depressive symptoms. The reliability of the PHQ9 questionnaire in this study, as measured by Cronbach's alpha coefficient, is 0.811.

## Measurement of physical activity

Individual physical activity was assessed using the Global Physical Activity Questionnaire (GPAQ), and PAQ was administered in-home using a computer assisted personal interviewing (CAPI) system by trained interviewers. The Metabolic Equivalent of Task (MET) value of different types of physical activity is different, and physical activity (PA) was calculated based on the frequency, duration, type of activity, and MET value of each PA per week using the following formula: PA (MET-min/wk) = MET × weekly frequency of each PA × duration, using the NHANES recommended MET values. PA classification was performed by categorizing subjects as PA ≥ 600 MET min/week and PA < 600 MET min/week using the recommended criteria from the American PA Guidelines [29].

## Statistical analysis

The data analysis methods include two parts: descriptive statistical analysis and network analysis. Firstly, this study used SPSS 26.0 to perform descriptive statistical analysis on all data, aiming to explore the role of basic information and demographic variables of the subjects. Secondly, this study employed R 4.3.3 software for network analysis to investigate the network structure relationships of depression symptoms under different levels of physical activity in various menopausal states. The steps of network analysis followed the standardized guidelines published by Epskamp et al [30], and the analysis content consisted of five parts: network estimation, visualization of the network, centrality index estimation, network comparison, and estimation of network accuracy and stability.

**Network estimation and visualization.** The r packages "BootNet" and "qgraph" were employed for network estimation and visualization [31]. Given that the Gaussian graphical model (GGM) is premised on the continuous normal distribution of data, it is noteworthy that the PHQ9 data in this study are uniformly right-skewed (S1 Table). This skewness can result in inaccurate estimations of the covariance matrix when using the GGM model. Furthermore, the GGM model exhibits sensitivity to extreme values, which could potentially lead to either an underestimation or overestimation of correlations between variables. For the construction of the network model, this study opted for the "Ising" algorithm, wherein circular nodes symbolize depressive symptoms and the connecting lines between nodes, referred to as edges, have thicknesses indicative of the magnitude of the partial correlation coefficient [32,33].Considering that the method of the Ising algorithm in modeling three-point items in network analysis is still under debate, this study re-coded all items of PHQ9 into binary categories, reporting "Not at all" to indicate the absence of depressive symptoms, and reporting "Several days", "More than half the days", "Nearly every day" to indicate the presence of depressive symptoms [20].

**Centrality index estimation.** In this study, the central position of network symptom nodes was scrutinized utilizing the concept of Strength centrality. Here, 'Strength' is defined as the cumulative connections of a specific symptom with other symptoms, thereby denoting the symptom's influence within the network [34].

**Network accuracy and stability estimation.** This study estimates the accuracy and stability of the network through the BootNet package in R language software [30]. The accuracy of edge weights in this study is estimated by the 95%

confidence interval of the whitened edge weights, the smaller the area covered by the confidence interval, the more accurate the edge estimation. Through the subset construction procedure, a certain proportion of subjects are deleted and the node centrality is re-estimated. When the correlation between this centrality and the original centrality index reaches 0.7, the proportion of deleted subjects is defined as the centrality stability coefficient (CS). When this coefficient is greater than 0.25, it represents that the stability has reached an acceptable range, and when this coefficient is greater than 0.5, it indicates good stability [30].

**Network comparison.** To compare the depression symptom networks across different menopausal levels and physical activity levels, this study adopted a permutation test method for network comparison from both global invariance and local invariance perspectives (1000 iterations). The network comparison analysis was performed using the NetworkComparisonTest package in R language software for both global invariance tests and local invariance tests [35]. The significance level was set at 0.05, and results less than 0.05 were considered to indicate significant differences. The global invariance test is divided into two parts [36]: the network structure invariance test and the network overall strength invariance test. The network structure invariance test explores the maximum difference in the absolute values of edge weights between networks, while the network overall strength invariance test examines the difference in the sum of the absolute values of all edge weights between networks. The local invariance test examines the differences in edge weights and node centrality indices within individual sample networks.

## Results

### Analysis of baseline data of the study population

The analysis of baseline differences among individuals with different levels of physical activity before and after menopause is shown in Table 1. The results indicate that there are 3755 female research participants in this study, of which 2179 are pre-menopausal women accounting for 58.03%, and 1576 are post-menopausal women accounting for 41.97%. Among pre-menopausal women, there is a significant difference in age among different levels of physical activity ($P=0.016$), with low physical activity individuals ($34.54\pm6.62$) being older than high physical activity individuals ($33.74\pm6.69$). Among post-menopausal women, there are significant differences in age and depressive symptoms (Anhedonia, Sleep, Energy) among different levels of physical activity, with low physical activity individuals ($62.20\pm11.85$) being older than high physical activity individuals ($60.78\pm12.56$), and the prevalence of depressive symptoms (Anhedonia, Sleep, Energy) is also higher in low physical activity individuals compared to high physical activity individuals.

### Network modelling of depressive symptoms in pre- and postmenopausal women

The network models representing depressive symptoms preceding and following menopause each contain 36 edges; however, the pre-menopausal model has 32 (88.9%) non-zero edges with an average weight of 0.659, while the post-menopausal model has 33 (83.3%) non-zero edges with an average weight of 0.649. The complexity of the pre-menopausal network is notably higher than that of the post-menopausal network, as depicted in Fig 1.

The centrality index estimation revealed that the PHQ02 (Sad Mood) exhibited the highest centrality, with values of $r_s=2.246$ and $r_s=2.063$ in the pre-menopausal and post-menopausal depression symptom network models, respectively. This suggests its pivotal role in the network, as illustrated in Fig 2. The centrality stability estimation results (S2 Fig) showed that, after placing most of the samples, the CS coefficient of centrality for nodes with menopausal symptoms was greater than 0.5, indicating overall good stability (S2A Fig); the CS coefficient of centrality for nodes with postmenopausal symptoms was greater than 0.75, indicating overall good stability (S2B Fig). The stability of centrality for nodes with postmenopausal symptoms was better than that of nodes with premenopausal symptoms. Analysis of the bootstrap results for edge weights in the depression symptom network, both pre- and post-menopause, reveals good accuracy (S3 Fig). The strongest association pre-menopause is observed between DP2 (Sad) and DP6 (Guilt) (S3A Fig), while post-menopause, the strongest association is between DP2 (Sad) and DP9 (Suicidal thoughts) (S3B Fig). A bootstrap

**Table 1. Analysis of differences in baseline data among pre- and postmenopausal women with different levels of physical activity.**

| Characteristic | Premenopausal (2179) | | P value | Postmenopausal (1576) | | P value |
|---|---|---|---|---|---|---|
| | PA<600(535) | PA≥600(1644) | | PA<600(573) | PA≥600(1003) | |
| RIDAGEYR (mean (SD)) | 34.54 (6.62) | 33.74 (6.69) | 0.016 | 62.20 (11.85) | 60.78 (12.56) | 0.027 |
| RIDRETH1 (%) | | | 0.061 | | | 0.738 |
| Mexican American | 95(17.8) | 281(17.1) | | 56(9.8) | 80(8.0) | |
| Non-Hispanic Black | 124(23.2) | 390(23.7) | | 144(25.1) | 246(24.5) | |
| Non-Hispanic White | 178(33.3) | 607(36.9) | | 272(47.5) | 501(50.0) | |
| Other Hispanic | 48(9.0) | 167(10.2) | | 49(8.6) | 89(8.9) | |
| Other Race – Including Multi-Racial | 90 (16.8) | 199(12.1) | | 52(9.1) | 87(8.7) | |
| DMDMARTL (%) | | | 0.091 | | | 0.104 |
| Married | 285 (53.3) | 806 (49.0) | | 254 (44.3) | 496 (49.5) | |
| Widowed | 7 (1.3) | 9 (0.5) | | 116 (20.2) | 180 (17.9) | |
| Divorced | 48 (9.0) | 138 (8.4) | | 129 (22.5) | 179 (17.8) | |
| Separated | 27 (5.0) | 79 (4.8) | | 22 (3.8) | 36 (3.6) | |
| Never married | 92 (17.2) | 360 (21.9) | | 34 (5.9) | 67 (6.7) | |
| Living with partner | 76 (14.2) | 252 (15.3) | | 18 (3.1) | 45 (4.5) | |
| Anhedonia(%) | | | 0.508 | | | 0.006 |
| asymptomatic | 393 (74.4) | 1197 (72.8) | | 370(64.6) | 716(71.4) | |
| symptomatic | 137 (25.6) | 447 (27.2) | | 203(35.4) | 287(28.6) | |
| Sad Mood (%) | | | 0.933 | | | 0.396 |
| asymptomatic | 386 (77.1) | 1181 (71.8) | | 392 (68.4) | 708 (70.6) | |
| symptomatic | 149 (27.9) | 463 (28.2) | | 181 (31.6) | 295 (29.4) | |
| Sleep(%) | | | 0.098 | | | 0.006 |
| asymptomatic | 335 (62.6) | 961 (55.9) | | 291 (50.8) | 582 (58.0) | |
| symptomatic | 200 (37.4) | 683 (41.5) | | 282 (49.2) | 421 (42.0) | |
| Energy(%) | | | 0.638 | | | 0.049 |
| asymptomatic | 218 (40.7) | 649 (39.5) | | 227 (39.6) | 450 (44.9) | |
| symptomatic | 317 (59.3) | 995 (60.5) | | 346 (60.4) | 553 (55.1) | |
| Appetite (%) | | | 0.146 | | | 0.426 |
| asymptomatic | 344 (64.3) | 1115 (67.8) | | 384 (67.0) | 693 (69.1) | |
| symptomatic | 191 (35.7) | 529 (32.2) | | 189 (33.0) | 310 (30.9) | |
| Guilt(%) | | | 0.777 | | | 0.791 |
| asymptomatic | 418 (78.1) | 1296 (78.8) | | 464 (81.0) | 819 (81.7) | |
| symptomatic | 117 (21.9) | 348 (21.2) | | 109 (19.0) | 184 (18.3) | |
| Concentration(%) | | | 0.626 | | | 0.656 |
| asymptomatic | 431 (80.6) | 1342 (81.6) | | 452 (78.9) | 802 (80.0) | |
| symptomatic | 104 (19.4) | 302 (18.4) | | 121 (21.1) | 201 (20.0) | |
| Motor (%) | | | 0.773 | | | 0.785 |
| asymptomatic | 475 (88.8) | 1450 (88.2) | | 500 (87.3) | 869 (86.6) | |
| symptomatic | 60 (11.2) | 194 (11.8) | | 73 (12.7) | 134 (13.4) | |
| Suicide thoughts(%) | | | 0.819 | | | 1.000 |
| asymptomatic | 516 (96.4) | 1591 (96.8) | | 549 (95.8) | 962 (95.9) | |
| symptomatic | 19 (3.6) | 53 (3.2) | | 24 (4.2) | 41 (4.1) | |

Note: RIDAGEYR is age; RIDRETH1 is race; DMDMARTL is marital status.

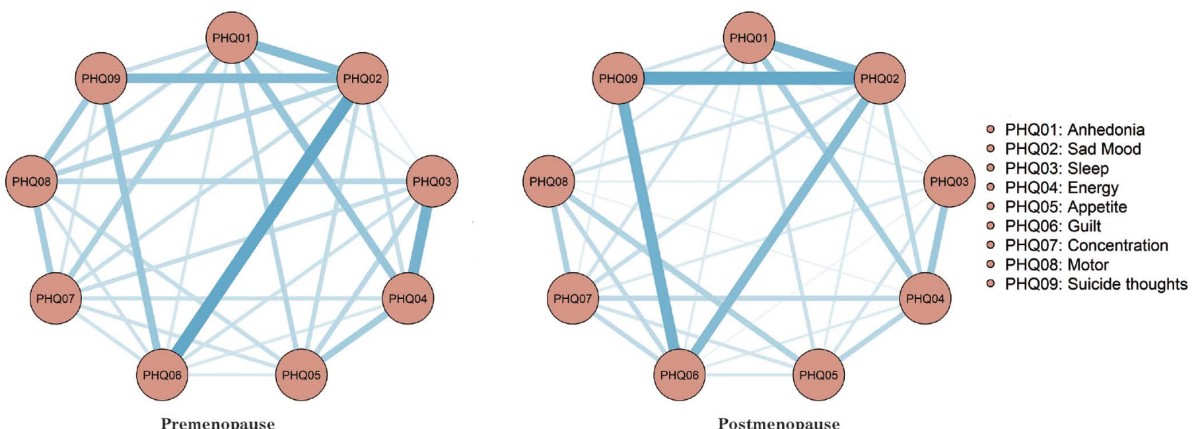

**Fig 1. Estimation of network model of dichotomous depressive symptoms before and after menopause.**

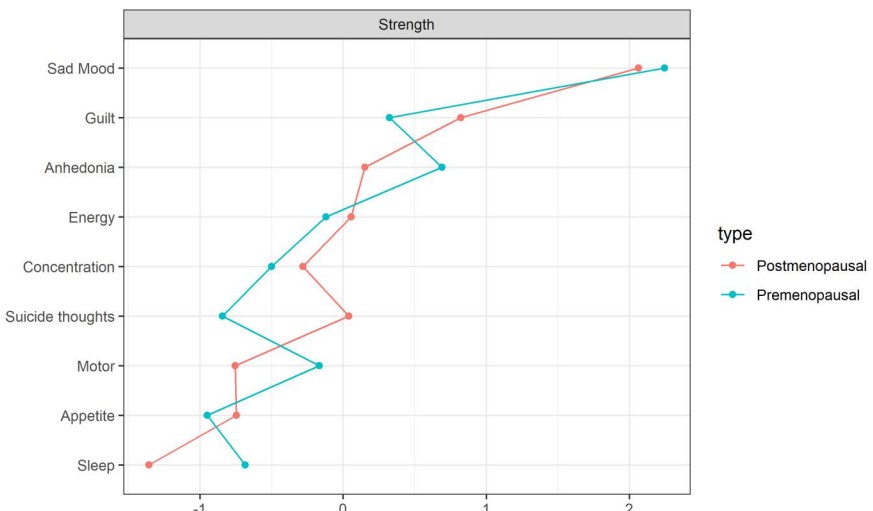

**Fig 2. Standardized estimation of centrality index.**

difference test estimating edge weight differences demonstrates that most edge weights are statistically significant ($p < 0.05$) (S4 Fig), with more statistically significant edge weights observed post-menopause (S4A Fig) compared to pre-menopause (S4B Fig).

We employed the Network Invariance Test and Global Strength Invariance Test to examine the differences between pre- and postmenopausal networks. The Network Invariance Test revealed no significant structural invariance among different menopausal groups: M = 2.337, P = 0.267. Similarly, the Global Strength Invariance Test indicated no notable difference: S = 1.705, P = 0.594. Furthermore, the edge invariance test demonstrated that none of the edges differed significantly between the networks.

The networks depicting depressive symptoms, both pre-menopausal and post-menopausal, exhibit high network accuracy and stability. In both networks, the symptom nodes of PHQ02 (Sad Mood, $r_s = 2.246$ and $r_s = 2.063$) maintain a central position and demonstrate close connectivity with other symptom nodes.

## Depression symptoms network models of different physical activities

In order to explore the differences and core symptoms of depression symptom network structure with different physical activities before and after menopause, this study also estimated the depression symptom network with two levels of different physical activities before and after menopause, respectively.

**Premenopausal different physical activity network models.** In the network analyzing depressive symptoms correlating with two levels of physical activity prior to menopause, both networks contain 36 edges. These yield 32 (88.9%) and 24 (66.7%) non-zero weighted edges, with average weights being 0.653 and 0.501 respectively. The network model representing physical activity less than 600 is notably more intricate than that depicting activity 600 or above. As illustrated in Fig 3, there are distinct consistencies, such as the close connection between PHQ02 (Sad Mood) and PHQ06 (Guilt), and discrepancies, notably the disappearance of a link between PHQ02 (Sad Mood) and PHQ09 (Suicide thoughts) as physical activity escalates, as detailed in Fig 3.

The estimation of the centrality index indicates that the PHQ02 (Sad Mood) symptom node exhibits the highest centrality within the depression symptom network across various physical activity levels, with rs values of 2.289 and 0.898 respectively. Notably, the PHQ02 (Sad Mood) symptom demonstrates a more pronounced effect in the network model where physical activity is less than 600 (Fig 4).The network centrality index stability estimation results indicate that both the PA<600 and PA≥600 depression network models exhibit considerable stability, as evidenced by a Strength CS coefficient exceeding 0.5 (S5 Fig). Edge weight bootstrapping results for both network models suggest commendable accuracy (S6 Fig). The most significant association edge in the PA<600 model is between nodes DP2 (Sad Mood) and DP6 (Guilt), and this association remains consistent in the PA≥600 model (S7A and S7B Fig). The bootstrap difference test results for estimating the variance in edge weight revealed (S7 Fig) that the majority of edge weights in the PA<600 network model were statistically significant (S7A Fig). Conversely, only a small fraction of edge weights in the PA≥600 network model demonstrated statistical significance (S7B Fig).

Upon comparing the networks of depressive symptoms across two distinct physical activities, analyses of network structure invariance and strength invariance reveal no significant differences in the depressive symptom network associated with varying physical activities (M=1.429, p=0.659; S =5.451, P=0.176). Moreover, the edge invariance test indicates no significant disparity between the two groups.

In various physical activity network models, the PA<600 network model, with 32 edges (representing 88.9% connectivity), exhibits a more intricate and interconnected structure than the PA≥600 network model, which has 24 edges

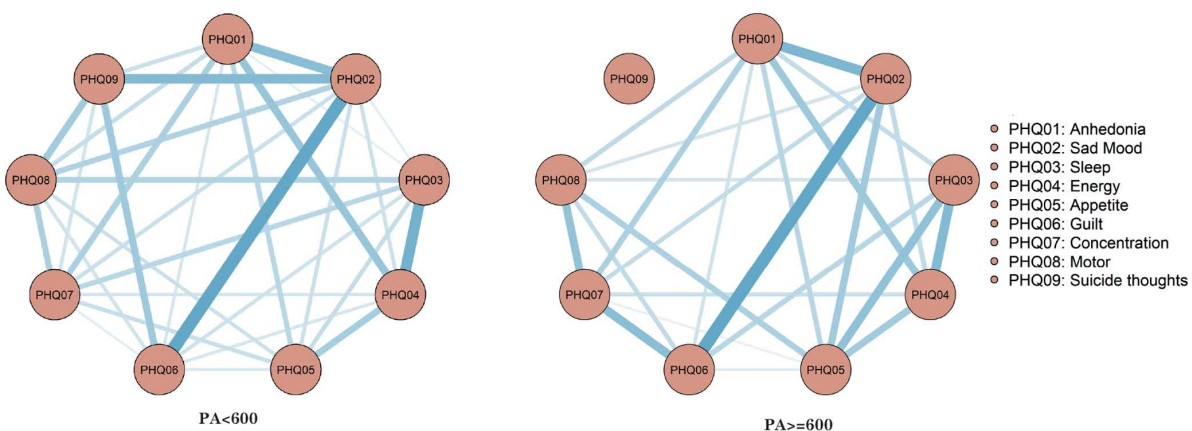

**Fig 3. Premenopausal depression symptom estimation network model for different physical activities.**

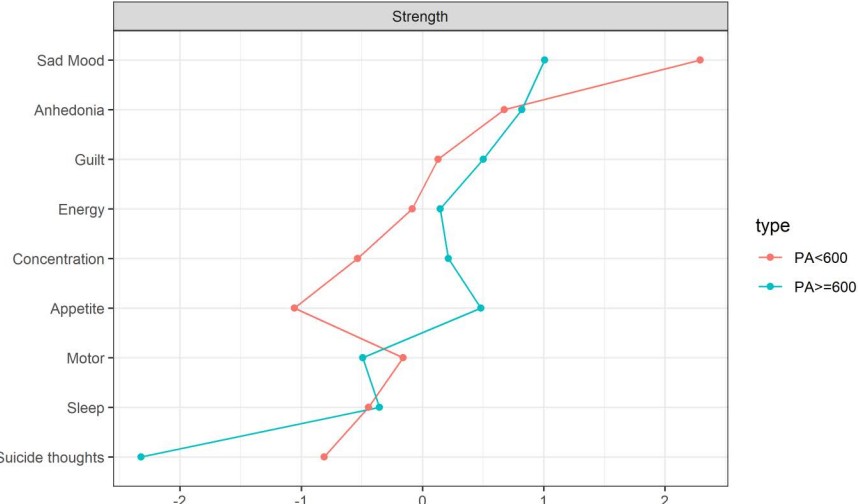

**Fig 4. Premenopausal standardized estimation of different physical activity network centrality indices.**

(representing 66.7% connectivity). Furthermore, the symptom node PHQ02, indicative of Sad Mood, demonstrates a stronger performance with an $r_s$ value of 2.289.

**Postmenopausal different physical activity network models.** In the networks analyzing depressive symptoms correlating with varying levels of physical activity post-menopause, each contains 36 edges, yielding 29 (80.5%) and 26 (72.2%) non-zero weight edges respectively, with average weights of 0.638 and 0.604. The network model representing physical activity levels less than 600 (PA<600) demonstrates greater complexity than that of physical activity levels equal to or exceeding 600 (PA≥600). There are noteworthy differences between the depressive symptom networks associated with these two levels of physical activity. Specifically, in the PA<600 network model, the connections between PHQ02 (Sad Mood) and PHQ01 (Anhedonia), PHQ06 (Guilt) diminish significantly in the PA≥600 network model, as illustrated in Fig 5. The centrality index estimation reveals that within the network of depression symptoms for those with PA<600, the PHQ01 (Sad mood) symptom node demonstrates the highest centrality with an rs=2.060. Conversely, in the network for those with PA≥600, the PHQ09 (Suicide thoughts) symptom node exhibits the most pronounced centrality with an rs=1.778, as depicted in Fig 6. An examination of the stability estimation results for the centrality index of the depression symptom network across different physical activity levels (S8 Fig) indicates that the network centrality index for PA<600 is CS>0.75, suggesting robust stability. However, for PA≥600, the network centrality index is CS<0.25 (S8A Fig), signaling compromised stability (S8B Fig).The edge weight bootstrapping outcomes for the network models at varying physical activity levels indicate commendable accuracy for both (S9 Fig). The most robust association in the PA<600 model is observed between DP2 (Sad) and DP9 (Suicide thoughts) (S9A Fig), whereas the strongest association in the PA≥600 model is between DP6 (Guilt) and DP9 (Suicide thoughts) (S9B Fig). Furthermore, edge weight difference estimations via bootstrapping difference tests demonstrate that a majority of the edge weights in the PA<600 network model are statistically significant (S10A Fig), whereas only a select few in the PA≥600 network model achieve statistical significance (S10B Fig).

Upon comparing the networks of depressive symptoms with low and high physical activity, it was observed that there were no significant differences in the network structure invariance and strength invariance. This indicates that the depressive symptom networks remain consistent regardless of the level of physical activity (M=2.000, P=0.369; S =2.868, P=0.531). Furthermore, there was no significant difference in the comparison between the two regarding the edge invariance test.

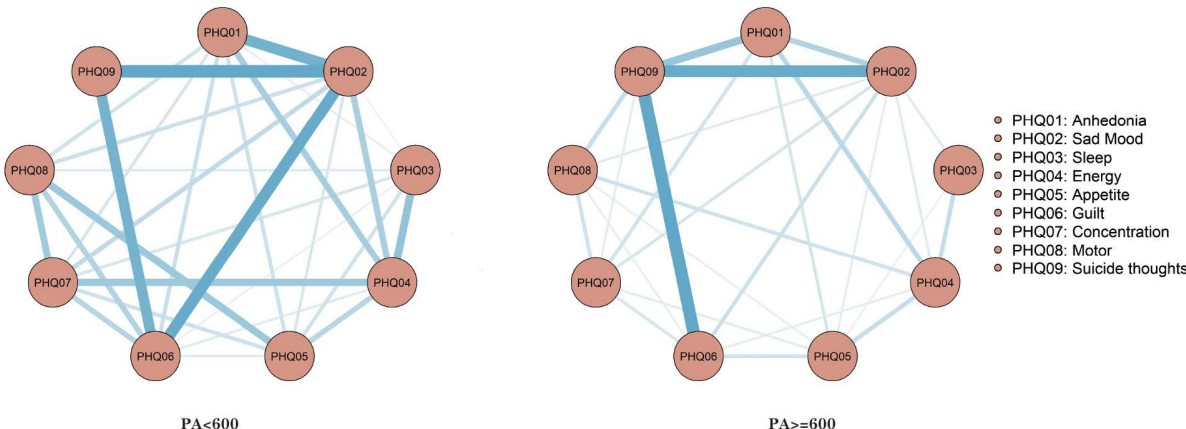

**Fig 5. Postmenopausal depression symptoms estimation network model for different physical activities.**

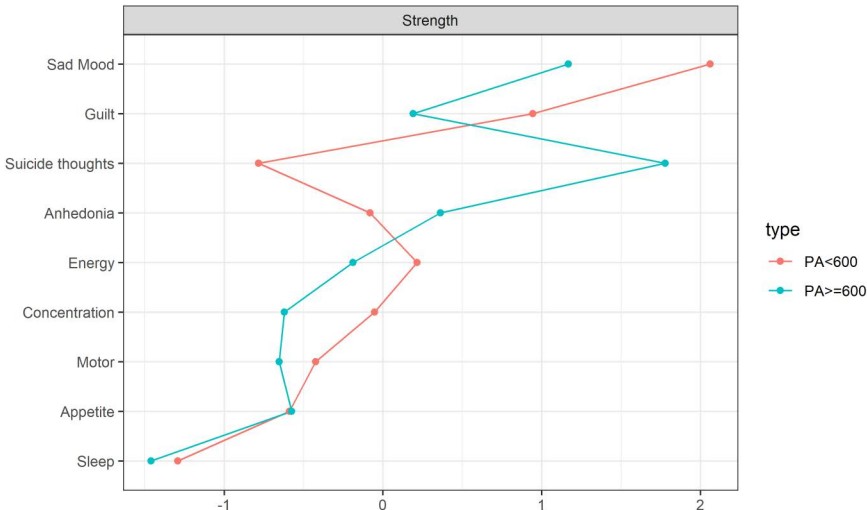

**Fig 6. Postmenopausal standardized estimation of different physical activity network centrality indices.**

In the network model analyzing postmenopausal depression symptoms in relation to various physical activities, the PA<600 network model, with 29 edges (representing 80.5% connectivity), demonstrates a more intricate and closely interconnected structure than the PA≥600 network model, which has 26 edges (signifying 72.2% connectivity). Within the PA<600 network, the symptom node labeled PHQ01 (Sad Mood, $r_s$=2.060) displays the most pronounced performance. Notably, the overall stability and accuracy of the PA<600 network surpass those of the PA≥600 network.

## GGM model network analysis

Compared with the GGM model network, the Ising network model has a higher similarity in terms of network edge size (r=0.935, 95% CI = [0.428; 0.668], P<0.001) and global strength (Ising=23.726, GGM=4.011, 95% CI = [0.539, 2.150], P<0.001) in premenopausal women; and node ranking r=0.933. In postmenopausal women, the Ising network model has a higher similarity in terms of network edge size (r=0.889, P<0.001) and global strength (Ising=21.804, GGM=3.854, P<0.001); and node ranking r=0.850. More details of the rest of the networks are shown in S2 and S3 Tables and S11 Fig.

**Covariates (age, education level, and marital status)**

Prior research has indicated a significant correlation between age, education level, and marital status with the epidemiological and clinical characteristics of Major Depressive Disorder (MDD). Consequently, these variables were controlled for as covariates in re-estimating the network model and local structure index. The resulting networks, both from the original model and the covariate-controlled model, exhibited near-identical structures in terms of edges and strength.

In premenopausal women, the crude network model was compared to the control covariate network model in terms of edge size (r = 0.889, 95% CI = [0.778; 0.955], P < 0.001) and strength (r = 0.747, 95% CI = [0.698; 0.930], P = 0.031). For postmenopausal women, the comparison revealed similar results for edge size (r = 0.858, 95% CI = [0.757; 0.950], P < 0.001) and strength (r = 0.860, 95% CI = [0.788; 0.986], P = 0.003). Further details on the remaining networks are provided in S4 Table and S12 Fig.

## Discussion

Through network analysis, we found that there are complex interactions among depressive symptoms. The network analysis results show that "Sad Mood" is the most core symptom in the network of depressive symptoms before and after menopause, which connects or triggers the remaining depressive symptoms in the network. This study also reveals different patterns of depressive symptoms between different levels of physical activity. The network model of depressive symptoms with high physical activity is looser and less stable than that with low physical activity, especially in the network of postmenopausal women. The centrality index of the depression network with high physical activity is unstable, and the network structure is sparser. This study shows that "Sad Mood" is the core symptom node in the network of depressive symptoms in women before and after menopause, a finding consistent with previous studies in the United States on psychiatric inpatients [37]. Notably, despite differences in sample characteristics, the stability of " Sad Mood " as a hallmark core symptom for depression diagnosis is confirmed across different populations. Further comparisons reveal that in the analysis of the PHQ-9 symptom network among the general German population and cancer patients, as well as in the network analysis of depressive symptoms among Hong Kong residents during the COVID-19 pandemic, " Sad Mood " is identified as the core symptom, similar to the findings of this study [20,38]. This suggests that, although menopausal women have a unique physiological-psychological interactive background (such as hormonal fluctuations), the pivotal position of " Sad Mood " may reflect the universal characteristics of the depressive symptom network across different populations. Neuroimaging research has demonstrated that sad mood is linked to structural and functional irregularities in the orbitofrontal cortex, medial prefrontal cortex [39–41], amygdala [41], and associated regions of the striatum and thalamus [42]. These areas typically govern limbic and brainstem structures implicated in the development of depressive symptoms [43]. Analysis of the causes of sadness from micro and macro perspectives, neurobiologically or from the standpoint of estrogen withdrawal theory, reveals that estrogen influences the metabolism of neurotransmitters such as dopamine, norepinephrine, β-endorphin, and serotonin, thereby affecting mood status [44]. From the bio-psycho-social-cultural model hypothesis viewpoint, life stress events during menopause (such as caring for parents or children's development) may heighten biological vulnerability [45]. Therefore, cognitive-behavioral therapy can be administered to premenopausal women to intervene or prevent the deterioration of their "sad mood". Moreover, the network comparison tes (NCT) results showed no significant differences in the structure of the depression symptom network between pre- and post-menopause, suggesting no distinct clinical features between these phases. A meta-analysis also found no increased likelihood of developing depression during perimenopause compared to pre-menopause, and no difference in the risk of depressive symptoms between pre-menopausal and post-menopausal phases [46].

We further revealed that the network model of depressive symptoms in the high-intensity physical activity was looser than that in the low-intensity physical activity both before and after menopause, which indicated that the intensity of physical activity affected the interconnection of depressive symptoms in women before and after menopause. This study also discovered that in the network model of PA < 600 both pre- and post-menopause, the symptom node of 'Sad Mood'

demonstrates the highest association strength in the network and serves as the central point for these symptoms. However, in the network model of PA≥600, the association strength of nodes representing a depressed mood diminishes, and the connection strength with each node weakens. This suggests that as physical activity levels increase, the correlation between depressive symptoms weakens, particularly in postmenopausal women. Accumulating evidence has also shown that physical activity is more closely related to positive emotions than negative emotions [47], and people with high levels of physical activity have a 17% lower chance of developing depression than those with low levels of physical activity [48,49]. The explanation for this association may be that physical activity can trigger acute neuroendocrine responses, such as the activation of the endogenous cannabinoid system and long-term adaptation of brain neurostructure changes, and physical activities such as running can increase the levels of endogenous cannabinoids, producing a sense of pleasure known as "runner's high"; from a psychosocial and behavioral perspective, physical activity can improve physical self-perception and body image, increase social interactions, and personal development of coping strategies [7,50]. Therefore, for postmenopausal women: risk of developing depression can be reduced using a combination of cognitive therapy and physical activity. Moreover, the results from the network comparison test revealed no significant differences in the invariance of network structure between the low and high physical activity (PA) groups. This implies that the pattern of depressive symptom expression and activation remains consistent across varying levels of physical activity both pre- and post-menopause. Notably, while high levels of physical activity are associated with a reduced likelihood of depression compared to low levels [49], there is a lack of evidence to suggest that the relationship model between depressive symptoms and physical activity levels varies.

While this study introduces a degree of novelty, it is not without limitations. The data is sourced from a cross-sectional study, which precludes the ability to discern whether depressive symptoms precipitated other observed phenomena or vice versa, thereby inhibiting causal inference. Furthermore, the study population exclusively comprises women aged over 20, limiting the generalizability of results to other demographics, such as minors or individuals with severe depression. Additionally, although the PHQ-9 is a psychometrically valid instrument that aligns with DSM-5 diagnostic criteria for major depression, its design, which amalgamates two symptom directions into a single bidirectional question (e.g., regarding appetite decrease or increase), while streamlining the questionnaire, may sacrifice nuanced details concerning specific symptomatic performances [22]. This aspect will be duly considered in future work, paving the way for more comprehensive research.

## Conclusions

This study, utilizing network analysis, discovered that the symptom "Sad Mood" assumes a pivotal role within the network of depressive symptoms both pre- and post-menopause. Furthermore, it was discerned that high levels of physical activity have the potential to diminish the significance of this symptom within the network, thereby attenuating its association with other symptoms. Therefore, it may be crucial to closely monitor the "Sad Mood" symptom in women during and after menopause and incorporate physical activity as a strategy to mitigate and forestall the exacerbation of depressive pain.

## Supporting information

**S1 File. Supplementary materials.** Contains 12 supplementary figures (S1–S12 Figs) and 4 supplementary tables (S1–S4 Tables) supporting the network analysis of depressive symptoms in pre- and postmenopausal women, including centrality stability, edge weight bootstrap tests, and model comparisons.
(ZIP)

## Author contributions

**Conceptualization:** Fashui Gao, Guofang Ma.

**Data curation:** Song Chen.

**Investigation:** Yuqi Tian, Ruixin Ma.

**Methodology:** Song Chen, Di Wu, Guofang Ma.

**Software:** Song Chen.

**Visualization:** Song Chen.

**Writing – original draft:** Song Chen, KaiYue Nie, Fashui Gao.

**Writing – review & editing:** Song Chen, Di Wu, KaiYue Nie, Yuqi Tian, Ruixin Ma, Guofang Ma.

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
