## [Decision Letter · Decision Letter 0]

PONE-D-25-01703Network Analysis of Depression Symptoms and Physical Activity Levels Before and After MenopausePLOS ONE

Dear Dr. Ma,

Thank you for submitting your manuscript to PLOS ONE. After careful consideration, we feel that it has merit but does not fully meet PLOS ONE’s publication criteria as it currently stands. Therefore, we invite you to submit a revised version of the manuscript that addresses the points raised during the review process.

Thank you for your scholarly contribution. I'd also like to thank the authors for choosing PLOS ONE to publish your findings from this study. Comments from reviewers are provided below. Please review these comments and I suggest address them and resubmit your manuscript. Your timely response would help this study be published and will make it accessible to interested readers across the world. I look forward to reviewing your revised manuscript. I wish you good luck with your future endeavors.

We look forward to receiving your revised manuscript.

Kind regards,

Lakshminarayana Chekuri, MD, PhD

Academic Editor

PLOS ONE

Journal Requirements:

2.Thank you for uploading your study's underlying data set. Unfortunately, the repository you have noted in your Data Availability statement does not qualify as an acceptable data repository according to PLOS's standards.

Additional Editor Comments:

Thank you for your scholarly contribution. I'd also like to thank the authors for choosing PLOS ONE to publish your findings from this study. Comments from reviewers are provided below. Please review these comments and I suggest address them and resubmit your manuscript. Your timely response would help this study be published and will make it accessible to interested readers across the world. I look forward to reviewing your revised manuscript. I wish you good luck with your future endeavors.

Reviewers' comments:

Reviewer's Responses to Questions

**Comments to the Author**

1. Is the manuscript technically sound, and do the data support the conclusions?

Reviewer #1: Partly

Reviewer #2: Partly

2. Has the statistical analysis been performed appropriately and rigorously? 

Reviewer #1: Yes

Reviewer #2: No

3. Have the authors made all data underlying the findings in their manuscript fully available?

Reviewer #1: Yes

Reviewer #2: Yes

4. Is the manuscript presented in an intelligible fashion and written in standard English?

Reviewer #1: Yes

Reviewer #2: Yes

5. Review Comments to the Author

Reviewer #1: “Network Analysis of Depression Symptoms and Physical Activity Levels Before and After Menopause”(PONE-D-25-01703)

This manuscript aimed to explore the depression symptoms according to physical activity level before and after menopause using network analysis and the data from NHANS. The results revealed that “Sad Mood' emerged as the most central symptom, positioning itself as the core of the network. Furthermore, there was a noticeable decrease in the correlation between depressive symptoms and a reduced stability in the network structure during periods of high physical activity compared to those of low physical activity. Notably, no significant structural differences were observed between the pre-menopausal and post-menopausal network models, regardless of physical activity levels. Overall, this topic is interesting and important. However, some concerns appeared after reading the whole manuscript.

1. Some important papers are missing and need to be reviewed and discussed, such as

Martin-Key, N. A., Funnell, E. L., Benacek, J., Spadaro, B., & Bahn, S. (2024). Using network analysis to understand the association between menopause and depressive symptoms. npj Women's Health, 2(1), 41.

Wen, J., Wang, W., Liu, K., Sun, X., Zhou, J., Hu, H., ... & Miao, M. (2024). The psychological side of menopause: evidence from the comorbidity network of menopausal, anxiety, and depressive symptoms. Menopause, 10-1097.

Cai, H., Chen, M. Y., Li, X. H., Zhang, L., Su, Z., Cheung, T., ... & Xiang, Y. T. (2024). A network model of depressive and anxiety symptoms: a statistical evaluation. Molecular psychiatry, 29(3), 767-781.

Fausto, D. Y., Leitao, A. E., Silveira, J., Martins, J. B., Dominski, F. H., & Guimaraes, A. C. (2023). An umbrella systematic review of the effect of physical exercise on mental health of women in menopause. Menopause, 30(2), 225-234.

Soares, C. N. (2019). Depression and menopause: an update on current knowledge and clinical management for this critical window. Medical Clinics, 103(4), 651-667.

For the paragraph following the introduction of network analysis in the introduction, it is important to review the current available evidence to explore the related topics using network analysis.

2.“These symptoms significantly impact society, affecting Manuscript approximately 350 million people globally.” This statement needs references.

3.Qgraph needs reference

Epskamp S, Cramer AOJ, Waldorp LJ, Schmittmann VD, Borsboom D. qgraph: network Visualizations of Relationships in Psychometric Data. J Stat Softw. 2012;48(4):1–18. doi:10.18637/jss.v048.i04

4. Since there are no length limits of plos one, thus, the important information in the supplementary material should be included in the formal context.

5. Please provide effect sizes where available.

6. The theoretical implications should be discussed in more details.

7. It should be noted that there are many demographic differences between Premenopausal and Postmenopausal groups and this is also the same case for PA<600 and PA≥600 groups were revealed. These should be included as Covariates in the analysis.

Reviewer #2: 1.Please provide reliability coefficients (e.g., Cronbach's alpha) to evaluate the internal consistency and stability of your collected data.

2.There are several formatting and typographical errors that need correction. For instance, why is "N" capitalized in "BootNet". Also, "BootNe" is missing the "t" in section 2.4.3. Similar formatting and spelling errors appear elsewhere in the manuscript.

3.In section 3.2, second paragraph, instead of using the qualitative statement "Sadness holds a central role," please specify which nodes exhibit the highest strength indices with their corresponding values. This would provide more precise and quantifiable results.

4.The use of the Ising model requires further justification. While you express concerns about modeling ordinal data, it should be noted that most network analyses of PHQ-9 utilize the GGM model, which has been confirmed as appropriate for this type of data by network analysis methodology developers.

5.I recommend reporting only the Strength centrality index rather than all three indices (Strength, Closeness, and Betweenness). Consequently, the discussion section should focus on interpreting findings based on Strength centrality. This would enhance the clarity and interpretability of your results.

6.In your comparison of physical activity networks between pre- and post-menopausal women, the NCT results indicate no significant differences. Your statement that "The network model of PA<600 is more complex than that of PA≥600" requires precise quantification. Please define network complexity using specific metrics in this context.

7.The manuscript requires comprehensive language editing, particularly regarding network analysis terminology. Please ensure consistent and precise usage of technical terms throughout the text.

Implementation of these revisions would substantially improve the methodological rigor and clarity of your manuscript.

6. PLOS authors have the option to publish the peer review history of their article (what does this mean? ). If published, this will include your full peer review and any attached files.

**Do you want your identity to be public for this peer review?** For information about this choice, including consent withdrawal, please see our Privacy Policy .

Reviewer #1: No

Reviewer #2: No

---

## [Author Response · Author response to Decision Letter 1]

9 Mar 2025

Dear Editor and reviewers :

Thank you very much for the professional advice given by the editor and review experts, we have made changes and additions to the issues raised by the editor and the reviewing experts.The specific modifications are as follows:

Response to reviewer 1:

1.Some important papers are missing and need to be reviewed and discussed, such as

Martin-Key, N. A., Funnell, E. L., Benacek, J., Spadaro, B., & Bahn, S. (2024). Using network analysis to understand the association between menopause and depressive symptoms. npj Women's Health, 2(1), 41.

Wen, J., Wang, W., Liu, K., Sun, X., Zhou, J., Hu, H., ... & Miao, M. (2024). The psychological side of menopause: evidence from the comorbidity network of menopausal, anxiety, and depressive symptoms. Menopause, 10-1097.

Cai, H., Chen, M. Y., Li, X. H., Zhang, L., Su, Z., Cheung, T., ... & Xiang, Y. T. (2024). A network model of depressive and anxiety symptoms: a statistical evaluation. Molecular psychiatry, 29(3), 767-781.

Fausto, D. Y., Leitao, A. E., Silveira, J., Martins, J. B., Dominski, F. H., & Guimaraes, A. C. (2023). An umbrella systematic review of the effect of physical exercise on mental health of women in menopause. Menopause, 30(2), 225-234.

Soares, C. N. (2019). Depression and menopause: an update on current knowledge and clinical management for this critical window. Medical Clinics, 103(4), 651-667.

For the paragraph following the introduction of network analysis in the introduction, it is important to review the current available evidence to explore the related topics using network analysis.

We sincerely appreciate the reviewer's insightful suggestion regarding the integration of network analysis literature. As recommended, we have substantially revised the introduction section to incorporate recent advancements in network analysis applied to menopausal mental health.

2.“These symptoms significantly impact society, affecting Manuscript approximately 350 million people globally.” This statement needs references.

We would like to thank the reviewers for their suggestions.We have revised it in the text.

3.Qgraph needs reference

Epskamp S, Cramer AOJ, Waldorp LJ, Schmittmann VD, Borsboom D. qgraph: network Visualizations of Relationships in Psychometric Data. J Stat Softw. 2012;48(4):1–18. doi:10.18637/jss.v048.i04

We sincerely appreciate the advice given by the reviewers and I have made changes in the article

4.Since there are no length limits of plos one, thus, the important information in the supplementary material should be included in the formal context.

We would like to thank the reviewers for their suggestions.We have revised it in the text.

5.Please provide effect sizes where available.

We would like to thank the reviewers for their suggestions.We have revised it in the text.

6.The theoretical implications should be discussed in more details.

We would like to thank the reviewers for their suggestions.We have revised it in the text.

7.It should be noted that there are many demographic differences between Premenopausal and Postmenopausal groups and this is also the same case for PA<600 and PA≥600 groups were revealed. These should be included as Covariates in the analysis.

We sincerely thank the reviewers for highlighting this critical methodological consideration. As suggested, we are adding this analysis to the final results section.

Response to reviewer 2:

1.1.Please provide reliability coefficients (e.g., Cronbach's alpha) to evaluate the internal consistency and stability of your collected data.

We sincerely thank the reviewers for their suggestions, which I have elaborated on in the Methods section PHQ9's Cronbach's alpha

2.There are several formatting and typographical errors that need correction. For instance, why is "N" capitalized in "BootNet". Also, "BootNe" is missing the "t" in section 2.4.3. Similar formatting and spelling errors appear elsewhere in the manuscript.

We would like to thank the reviewers for their suggestions. “BootNet” is a program package for the r-language, which I have modified in the text.

3.In section 3.2, second paragraph, instead of using the qualitative statement "Sadness holds a central role," please specify which nodes exhibit the highest strength indices with their corresponding values. This would provide more precise and quantifiable results.

We would like to thank the reviewers for their suggestions.We have revised it in the text.

4.The use of the Ising model requires further justification. While you express concerns about modeling ordinal data, it should be noted that most network analyses of PHQ-9 utilize the GGM model, which has been confirmed as appropriate for this type of data by network analysis methodology developers.

We would like to thank the reviewers for their suggestions.Given that the Gaussian graphical model (GGM) is premised on the continuous normal distribution of data, it is noteworthy that the PHQ9 data in this study are uniformly right-skewed (Supplementary Table 1). This skewness can result in inaccurate estimations of the covariance matrix when using the GGM model. Furthermore, the GGM model exhibits sensitivity to extreme values, which could potentially lead to either an underestimation or overestimation of correlations between variables. For the construction of the network model, this study opted for the "Ising" algorithm, wherein circular nodes symbolize depressive symptoms and the connecting lines between nodes, referred to as edges, have thicknesses indicative of the magnitude of the partial correlation coefficient.

5.I recommend reporting only the Strength centrality index rather than all three indices (Strength, Closeness, and Betweenness). Consequently, the discussion section should focus on interpreting findings based on Strength centrality. This would enhance the clarity and interpretability of your results.

We would like to thank the reviewers for their suggestions.We have revised it in the text.

6.In your comparison of physical activity networks between pre- and post-menopausal women, the NCT results indicate no significant differences. Your statement that "The network model of PA<600 is more complex than that of PA≥600" requires precise

We would like to thank the reviewers for their suggestions.We have revised it in the text.

7.The manuscript requires comprehensive language editing, particularly regarding network analysis terminology. Please ensure consistent and precise usage of technical terms throughout the text.

We would like to thank the reviewers for their suggestions.We have revised it in the text.

We have also made changes to other inappropriate parts of the original manuscript, including but not limited to: language presentation, number of references, and writing error issues. All of the above changes are highlighted in red in the article.

We are very grateful to the editors and reviewers for their careful guidance. Due to the limitation of academic level, if there are any inappropriate or imprecise areas, we hope you can give us your advice, and we will revise them carefully in time.

Sincerely yours,

Song Chen

---

## [Decision Letter · Decision Letter 1]

PONE-D-25-01703R1Network Analysis of Depression Symptoms and Physical Activity Levels Before and After MenopausePLOS ONE

Dear Dr. Ma,

Thank you for submitting your manuscript to PLOS ONE. After careful consideration, we feel that it has merit but does not fully meet PLOS ONE’s publication criteria as it currently stands. Therefore, we invite you to submit a revised version of the manuscript that addresses the points raised during the review process.

We look forward to receiving your revised manuscript.

Kind regards,

Lakshminarayana Chekuri, MD, PhD

Academic Editor

PLOS ONE

Journal Requirements:

Additional Editor Comments:

Thank you for your scholarly contribution. I'd also like to thank the authors for choosing PLOS ONE to publish your findings from this study. Comments from reviewers are provided below. Please review these comments and I suggest address them and resubmit your manuscript. Your timely response would help this study be published and will make it accessible to interested readers across the world. I look forward to reviewing your revised manuscript. I wish you good luck with your future endeavors.

Reviewers' comments:

Reviewer's Responses to Questions

**Comments to the Author**

1. If the authors have adequately addressed your comments raised in a previous round of review and you feel that this manuscript is now acceptable for publication, you may indicate that here to bypass the “Comments to the Author” section, enter your conflict of interest statement in the “Confidential to Editor” section, and submit your "Accept" recommendation.

Reviewer #1: All comments have been addressed

Reviewer #2: All comments have been addressed

2. Is the manuscript technically sound, and do the data support the conclusions?

Reviewer #1: Yes

Reviewer #2: Yes

3. Has the statistical analysis been performed appropriately and rigorously? 

Reviewer #1: Yes

Reviewer #2: No

4. Have the authors made all data underlying the findings in their manuscript fully available?

Reviewer #1: Yes

Reviewer #2: Yes

5. Is the manuscript presented in an intelligible fashion and written in standard English?

Reviewer #1: Yes

Reviewer #2: Yes

6. Review Comments to the Author

Reviewer #1: (No Response)

Reviewer #2: Dear Editor:

Thank you for forwarding the authors' revised manuscript for my review. I believe the authors have thoughtfully addressed most of the concerns I previously raised; however, there are still a few aspects that require further refinement:

Use of the Ising model: While the authors have explained their rationale for choosing the Ising model over the GGM model, this justification needs strengthening. The skewed distribution of data is typically not sufficient justification for dichotomizing data and employing the Ising model. I suggest that the authors:

Provide specific references supporting this methodological choice

Or, alternatively, consider conducting analyses using the GGM model as well, and comparing the results from both methods

This would enhance the robustness of the research methodology and provide readers with a more comprehensive analytical perspective.

Inconsistent figure citation format: Throughout the manuscript, there is inconsistency in how figures are referenced. Some instances use "Figure 3" while others use "Fig. 1" or "Fig. 2". I recommend adopting a uniform citation format, preferably following PLOS ONE's formatting guidelines. This consistency would improve the professionalism and readability of the article.

Inconsistent p-value reporting: The reporting of p-values in the text lacks uniformity, with some values presented in the exact format "p=0.016" and others in the range format "P<0.05". I suggest the authors standardize the p-value reporting format and ensure consistency in capitalization (either consistently using lowercase "p" or uppercase "P").

Apart from these points, the revised manuscript shows significant improvement in quality. If the authors can address these remaining issues, I believe the paper will be more complete and suitable for publication in PLOS ONE.

7. PLOS authors have the option to publish the peer review history of their article (what does this mean? ). If published, this will include your full peer review and any attached files.

**Do you want your identity to be public for this peer review?** For information about this choice, including consent withdrawal, please see our Privacy Policy .

Reviewer #1: No

Reviewer #2: No

---

## [Author Response · Author response to Decision Letter 2]

24 Apr 2025

Dear Editor and reviewers :

Thank you very much for the professional advice given by the editor and review experts, we have made changes and additions to the issues raised by the editor and the reviewing experts.The specific modifications are as follows:

Response to reviewer #2:

1.Use of the Ising model: While the authors have explained their rationale for choosing the Ising model over the GGM model, this justification needs strengthening. The skewed distribution of data is typically not sufficient justification for dichotomizing data and employing the Ising model. I suggest that the authors:

Provide specific references supporting this methodological choice

Or, alternatively, consider conducting analyses using the GGM model as well, and comparing the results from both methods

This would enhance the robustness of the research methodology and provide readers with a more comprehensive analytical perspective.

Thank you very much for the suggestions you made, and we fully recognize the importance of transparency and robustness in model selection for the credibility of the study's conclusions. In response to your comments, we have revised the manuscript.

2.Inconsistent figure citation format: Throughout the manuscript, there is inconsistency in how figures are referenced. Some instances use "Figure 3" while others use "Fig. 1" or "Fig. 2". I recommend adopting a uniform citation format, preferably following PLOS ONE's formatting guidelines. This consistency would improve the professionalism and readability of the article.

Inconsistent p-value reporting: The reporting of p-values in the text lacks uniformity, with some values presented in the exact format "p=0.016" and others in the range format "P<0.05". I suggest the authors standardize the p-value reporting format and ensure consistency in capitalization (either consistently using lowercase "p" or uppercase "P").

We would like to thank the reviewers for their suggestions.We have revised it in the text.

We have also made changes to other inappropriate parts of the original manuscript, including but not limited to: language presentation, number of references, and writing error issues. All of the above changes are highlighted in red in the article.

We are very grateful to the editors and reviewers for their careful guidance. Due to the limitation of academic level, if there are any inappropriate or imprecise areas, we hope you can give us your advice, and we will revise them carefully in time.

Sincerely yours,

Song Chen

---

## [Decision Letter · Decision Letter 2]

PONE-D-25-01703R2Network Analysis of Depression Symptoms and Physical Activity Levels Before and After MenopausePLOS ONE

Dear Dr. Ma,

Thank you for submitting your manuscript to PLOS ONE. After careful consideration, we feel that it has merit but does not fully meet PLOS ONE’s publication criteria as it currently stands. Therefore, we invite you to submit a revised version of the manuscript that addresses the points raised during the review process.

We look forward to receiving your revised manuscript.

Kind regards,

Lakshminarayana Chekuri, MD, PhD

Academic Editor

PLOS ONE

Journal Requirements:

Additional Editor Comments:

Dear Authors,

I believe this manuscript is close to acceptance. Please address following minor corrections:

Line 141, page 3, please expand "MET" at first occurrence

Line 142, page 3, expand "PA" at first occurrence

Lines 144-145. page 3: Please consider rephrasing this sentence: "Physically active, while the remaining subjects were divided into low-intensity PA and high-intensity PA based on PA."

In Table 1: Please expand "RIDAGEYR" "RIDRETH1" and "DMDMARTL" as a foot note.

Line 426, page 14: Please expand "NCT" and abbreviate in parenthesis.

Reviewers' comments:

Reviewer's Responses to Questions

**Comments to the Author**

1. If the authors have adequately addressed your comments raised in a previous round of review and you feel that this manuscript is now acceptable for publication, you may indicate that here to bypass the “Comments to the Author” section, enter your conflict of interest statement in the “Confidential to Editor” section, and submit your "Accept" recommendation.

Reviewer #2: All comments have been addressed

2. Is the manuscript technically sound, and do the data support the conclusions?

Reviewer #2: Yes

3. Has the statistical analysis been performed appropriately and rigorously? 

Reviewer #2: Yes

4. Have the authors made all data underlying the findings in their manuscript fully available?

Reviewer #2: Yes

5. Is the manuscript presented in an intelligible fashion and written in standard English?

Reviewer #2: Yes

6. Review Comments to the Author

Reviewer #2: I appreciate the authors' reasonable responses to my previous concerns. All the issues I raised have been adequately addressed, and I find the revisions satisfactory. I have no further questions or comments at this time,

7. PLOS authors have the option to publish the peer review history of their article (what does this mean? ). If published, this will include your full peer review and any attached files.

**Do you want your identity to be public for this peer review?** For information about this choice, including consent withdrawal, please see our Privacy Policy .

Reviewer #2: No

---

## [Author Response · Author response to Decision Letter 3]

15 May 2025

Dear Editor :

Thank you very much for the professional advice given by the editor experts, we have made changes and additions to the issues raised by the editor experts.The specific modifications are as follows:

Response to Editor Comments:

Line 141, page 3, please expand "MET" at first occurrence

Line 142, page 3, expand "PA" at first occurrence

Lines 144-145. page 3: Please consider rephrasing this sentence: "Physically active, while the remaining subjects were divided into low-intensity PA and high-intensity PA based on PA."

In Table 1: Please expand "RIDAGEYR" "RIDRETH1" and "DMDMARTL" as a foot note.

Line 426, page 14: Please expand "NCT" and abbreviate in parenthesis.

We would like to thank the editor for their suggestions.We have revised it in the text.

We have also made changes to other inappropriate parts of the original manuscript, including but not limited to: language presentation, number of references, and writing error issues. All of the above changes are highlighted in red in the article.

We are very grateful to the editors and reviewers for their careful guidance. Due to the limitation of academic level, if there are any inappropriate or imprecise areas, we hope you can give us your advice, and we will revise them carefully in time.

Sincerely yours,

Song Chen

---

## [Editor Report · Decision Letter 3]

Network Analysis of Depression Symptoms and Physical Activity Levels Before and After Menopause

PONE-D-25-01703R3

Dear Dr. Ma,

We’re pleased to inform you that your manuscript has been judged scientifically suitable for publication and will be formally accepted for publication once it meets all outstanding technical requirements.

Kind regards,

Lakshminarayana Chekuri, MD, PhD

Academic Editor

PLOS ONE
---

## [Editor Report · Acceptance letter]

PONE-D-25-01703R3

PLOS ONE

Dear Dr. Ma,

I'm pleased to inform you that your manuscript has been deemed suitable for publication in PLOS ONE. Congratulations! Your manuscript is now being handed over to our production team.

Kind regards,

on behalf of

Dr. Lakshminarayana Chekuri

Academic Editor

PLOS ONE